# A Data-Driven Solution for the Cold Start Problem in Biomedical Image Classification

## Abstract

The demand for large quantities of high-quality annotated images poses a significant bottleneck for developing an effective deep learning-based classifiers in the biomedical domain. We present a simple yet powerful solution to the cold start problem, i.e., selecting the most informative data for annotation within unlabeled datasets. Our framework encompasses three key components: (i) Pretraining an encoder using self-supervised learning to construct a meaningful data representation of unlabeled data, (ii) sampling the most informative data points for annotation, and (iii) initializing a model ensemble to overcome the lack of validation data in such contexts. We test our approach on four challenging public biomedical datasets. Our strategy outperforms the state-of-the-art in all datasets and achieves a 7% improvement on leukemia blood cell classification task with 8 times faster performance. Our work facilitates the application of deep learning-based classifiers in the biomedical fields, offering a practical and efficient solution to the challenges associated with tedious and costly, high-quality data annotation.

## 1 Introduction

When collaborating with clinical or biomedical experts in the development of health AI models, computer scientists often encounter a fundamental question: "How many labels are required to train an accurate classifier?"

The central challenge revolves around the selection of initial data for annotation when no initial labels are available—a common conundrum known as the *cold start problem*. The cold start problem refers to the initial phase of training where, in the absence of any labels or prior knowledge about the data, we are tasked with identifying and selecting the most informative data points for annotation, a crucial step that lays the groundwork for any subsequent semi-supervised or fully supervised training. This is especially critical in the biomedical domain. The scarcity of expert time for manual annotation makes the cold start problem even more daunting, as it becomes a bottleneck in advancing medical AI applications (Yakimovich et al., 2021). Specifically in active learning and few-shot learning paradigms, previous works by Shetab Boushehri et al. (2022), Yi et al. (2022), and Jin et al. (2022) demonstrated that careful selection of the initial annotation budget significantly accelerates and facilitates reaching peak performance in models trained on biomedical images with limited annotations like few-shot learning and active learning. Biomedical images significantly differ from natural images in color, contrast, complexity, and class distribution (van der Plas et al., 2019). Respective datasets exhibit class imbalance, limited diversity in shapes and color ranges, and rely on subtle feature variations for class distinctions—characteristics not commonly found in natural images. Moreover, biomedical images vary significantly across domains and experimental setups, further complicating the analysis (Blasi et al., 2016; Zhou, 2018; Konz et al., 2022). Furthermore, the absence of a validation set and limited knowledge about class distribution and data imbalances during initial data selection pose additional challenges.

The cold start problem has recently drawn considerable attention, underscoring the necessity of developing advanced techniques capable of identifying a high-quality initial annotated subset (Chandra et al., 2021; Jin et al., 2022; Yi et al., 2022; Wang et al., 2022; Mannix & Bondell, 2023). All works so far acknowledge that unlabeled pretraining is beneficial in arranging a clustered latent that is more straight forward for sampling initial data (Chandra et al., 2021; Yi et al., 2022) which is widely explored by Bengar et al. (2021) in active learning concepts. Current approaches seek to identify an

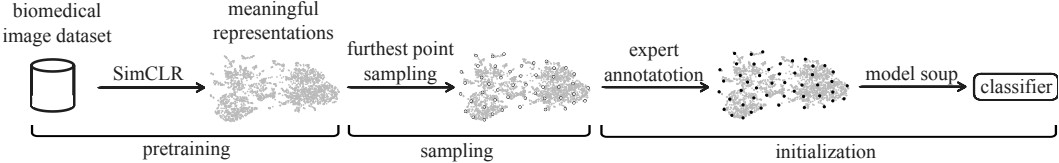

Figure 1: Our proposed framework has three steps for addressing the cold start problem for biomedical image classifiers: We employ SimCLR to pretrain the encoder and generate a meaningful representation of the unlabeled data. We apply furthest point sampling (FPS) to identify the most informative initial data to be labeled. Having a fixed budget for expert annotations for our budget, we start training the classifier head, where we apply model soups to achieve the best performance of the classifier in the absence of an adequate validation set.

informative annotation budget by sampling from latent space regions, either dense or sparse, using clustering and density estimation (Chandra et al., 2021; Jin et al., 2022). However, these methods haven't substantially outperformed random selection. This may be due to the sensitivity of many clustering techniques to parameters and dependency on prior knowledge about the class distribution of the data, while still, an accurate estimation of density in high (e.g., 128)-dimensional spaces is not guaranteed (Aggarwal et al., 2001). Some alternative methods propose adding optimizable clustering-based techniques to identify samples for the deep pre-trained encoder, providing more diverse samples based on the model's understanding (Wang et al., 2022; Mannix & Bondell, 2023). While promising, these techniques require significant resource and time during the initial training.

So far, none of the previous studies have applied their methods to the biomedical domain, where the cold start problem is both a practical concern and of significant importance. This highlights a notable research gap in addressing the unique challenges posed by biomedical datasets characterized by their complexity and the lack of comprehensive annotations.

We propose a straightforward solution for the cold start problem and test it on four biomedical image datasets. Building upon prior findings, we investigate three state-of-the-art self supervised learning (SSL) methods as a pretraining step to embed the entire unlabeled dataset in a meaningful latent space. Subsequently, we explore four different sampling strategies to select the most informative initial data points given a fixed annotation budget. Eventually, we address the lack of a validation set with model soups. Figure 1 depicts our proposed framework.

The main contributions of our works are:

- We are the first to address the cold start problem on challenging real-world biomedical datasets.
- We quantitatively compare three state-of-the-art self-supervised learning (SSL) methods—SimCLR (Chen et al., 2020), DINO (Caron et al., 2021), and SwAV (Caron et al., 2020)—to derive a meaningful representation of unlabeled data. We find SimCLR as the best SSL technique for biomedical data.
- We conduct a rigorous ablation study to assess the performance of four sampling strategies and identify furthest point sampling (FPS) (Qi et al., 2017) as the most effective technique to identify the most representative biomedical data points.
- We are the first proposing the model soups technique (Wortsman et al., 2022) to alleviate the challenges of lacking a reliable validation set and knowledge about classes distributions.
- We make our framework's code publicly available in a well documented repository, promoting transparency and reproducibility in research.

## 2 METHODDOLOGY

We begin with the dataset $X$ containing a total of $n$ images. Within this dataset, we define an annotation budget denoted as $(\tilde{X}, \tilde{Y})$. Here, $\tilde{X}$ represents a subset of $m$ images selected from $X$ (where $m \ll n$). This subset also includes corresponding labels denoted as $\tilde{Y}$, which are to be annotated by domain experts. This annotated budget, $(\tilde{X}, \tilde{Y})$, serves as the sole source of information for training a classifier model, denoted as $h_\gamma(f_\theta(.))$. This model comprises two main components: (i) Backbone

$f_\theta(.)$ with parameters $\theta$ that is responsible for encoding input images into a lower-dimensional latent space, denoted as $Z = f_\theta(X)$, and (ii) a linear classifier head $h_\gamma$, which takes the latent representations $Z$ as input and performs a classification tasks based on the provided labels $\tilde{Y}$. The classifier's parameters are denoted as $\gamma$.

**Pretraining.** The central challenge lies in the selection of informative and representative data for annotation where no information about labels and distribution of classes is provided. To address this challenge, we leverage the intrinsic information of the data through the self-supervised pretraining of the backbone $f_\theta(.)$. We consider SimCLR (Chen et al., 2020), SwAV (Caron et al., 2020), and DINO (Caron et al., 2021) architectures by embedding $f_\theta(.)$ as the deep encoder. These architectures show state-of-the-art performance in contrastive-instance, clustering-based, and self-distillation-based SSL approaches, respectively, and have demonstrated promising performance on widely recognized computer vision benchmark datasets, such as ImageNet (Russakovsky et al., 2015). At the end of pretraining, the trained backbone generates a meaningful latent representation of data, where semantically similar data are mapped close to each other and far from dissimilar data, resembling a clustered space ( see Figure 1).

**Sampling.** Random data selection lacks a strategic approach, treating all data points uniformly regardless of their information content or location within the latent space. This can results in annotating closely clustered or redundant data points while overlooking those at cluster boundaries, missing the opportunity to enhance model performance. Inspired by Qi et al. (2017), who used the FPS algorithm (see Algorithm 1) to sample points from non-uniform distributions within 3D object point clouds, we sample from the non-uniform distribution within our latent data space.

---

**Algorithm 1** Furthest point sampling (FPS)

---

1: $Z := \{z_1, \ldots, z_m\}$            // Set of all the points
2: $d_Z : Z \times Z \to \mathbb{R}_{\geq 0}$          // Distance metric
3: $m \in \mathbb{N}^+$                  // Number of samples
4: $\tilde{Z} \leftarrow \{z \in Z\}$            // Initialize the sampled points set with a random point
5: **while** $|\tilde{Z}| < m$ **do**
6:      $z^* = \arg \max_{z \in Z} \min_{\tilde{z} \in \tilde{Z}} d_Z(z, \tilde{z})$      // Furthest point from the sampled points set
7:      $\tilde{Z} \leftarrow \tilde{Z} \cup z^*$            // Update the sampled points set
8: **end while**
9: **return** $\tilde{Z}$

---

In the latent representation of our dataset $Z := \{z_1, z_2, ..., z_n\}$ FPS selects the first point randomly and then iteratively choses points $z^*$, in a way that maximizes the minimum distance to any of the previously selected points, i.e., $z^* = \arg \max_{z \in Z} \min_{\tilde{z} \in \tilde{Z}} D(z, \tilde{z})$, where $z^*$ is the selected point in the current iteration, $z$ represents a point in the point cloud $Z$, $\tilde{Z}$ is the set of points selected in previous iterations, and $D(z, \tilde{z})$ calculates the Euclidean distance between points $z$ and $\tilde{z}$. This method ensures the creation of a representative and well-distributed initial annotation set, effectively capturing both dense and sparse clusters within the data distribution. This systematic process guarantees that each newly chosen point contributes significantly to covering the remaining unselected data points, thus preserving the diversity of the data distribution mentioned by Wang & Ji (2020).

We also leverage the k-means clustering technique, known for its efficacy in high-dimensional space (Aggarwal et al., 2001). By applying k-means to the latent point cloud of unlabeled data, we aim to identify meaningful clusters. Subsequently, we employ three distinct sampling strategies: selecting data points closest to the centroids, opting for those farthest from the centroids, and a combination of half from the closest group and half from the farthest (closest/farthest). Given the absence of prior knowledge regarding the optimal number of clusters, we rigorously experiment with various k values to comprehensively explore the latent space's structure.

**Initialization.** We train the classifier head $h_\gamma$ by encoded sampled image set $\tilde{Z}$ and its corresponding labels $\tilde{Y}$ in a supervised manner. As conventional training-validation splits may not provide reliable results when annotated samples are scarce, we employ the concept of "model soups" proposed by Wortsman et al. (2022). Model soups involve averaging the weights of multiple models, each trained

with varying hyperparameters. In our case, we focus on varying learning rates. This approach effectively bypasses the need for extensive validation, achieving comparable results. The principle behind this effectiveness lies in the exploration of global minima. Assuming the existence of a global minimum, different hyperparameters lead to classifiers with weights localized around this minimum (see Algorithm 2). We can effectively reduce noise and enhance model robustness by obtaining multiple sets of weights through varied hyperparameters and averaging them.

---

**Algorithm 2** Uniform model soup

---

1: $W_i := \{w_{i1}, \ldots, w_{in}\}$                // Weights of model $h_\gamma$
2: $\mathbf{W} := \{W_1, \ldots, W_m\}$            // Set of weights of all the trained models
3: $W^* = \left\{ \frac{1}{|\mathbf{W}|} \sum_{j=1}^n w_{1j}, \ldots, \frac{1}{|\mathbf{W}|} \sum_{j=1}^n w_{mj} \right\}$   // Averaging weights to make the model soup
4: **return** $W^*$

---

## 3 RELATED WORKS

**Cold start learning.** Chandra et al. (2021) delved into an exploration of label-free pretraining techniques, including SimCLR (Chen et al., 2020) for self-supervised learning (SSL), VAE (Kingma & Welling, 2013), and SCAN (Van Gansbeke et al., 2020) for unsupervised methods. These works revealed SimCLR as the most promising pretraining approach among these methods. On top of SSL pretraining, (Chandra et al., 2021; Jin et al., 2022; Yi et al., 2022; Mannix & Bondell, 2023) introduced unique sampling techniques to identify the most informative data. Jin et al. (2022) introduced a hierarchical clustering technique, to sample from high-density regions of latent space. While dense regions primarily capture nearby data points and may overlook those situated farther away, Yi et al. (2022) proposed monitoring the SSL model loss to select both challenging and straightforward data samples, achieving more balanced coverage. However, none of these methods show a significant improvement compared to random initial data selection. Recent studies by Wang et al. (2022) and Mannix & Bondell (2023) demonstrated significant performance improvements through the use of semi-supervised learning techniques. Wang et al. (2022), introduce an adaptable clustering-based methodology designed to pinpoint data points situated at the periphery of densely clustered regions within the latent space. These identified points are considered as the most informative candidates for forming the initial annotation budget. Mannix & Bondell (2023) utilize k-medoids sampling within the low-dimensional projection of the latent space, referred to as cold PAWS, using t-SNE (Mannix & Bondell, 2023). Their approach demonstrates improved time efficiency and superior performance compared to (Wang et al., 2022). However, t-SNE operates as a stochastic algorithm, resulting in varying outcomes across different runs. This inherent stochasticity can introduce unpredictability into the sampling process. Furthermore, it demands significant computational resources due to its computational intensity and sensitivity to hyperparameters. Consequently, these factors can significantly impact efficiency, particularly when handling large datasets. The primary limitation in most of previous studies is the oversight of the unavailability of a validation set, which can be seen as a form of information leakage in this context. None of the previous works have reported results that specifically address a real-world scenario, particularly one involving biomedical datasets, characterized by data complexity and the overall information about the dataset, making it a critical area for further research.

**Self-supervised pretraining.** The foundation of SimCLR, SwAV, and DINO lies in the pretraining of a deep encoder, denoted as $f_\theta(.)$, which serves as the backbone for SSL. Within the latent realm of SimCLR, data points representing meaningful features exhibit a distinctive tendency to cluster naturally along the surface of a hypersphere. This intrinsic clustering plays a pivotal role in defining the latent space's character. Consequently, the constructed latent point cloud ($Z$) in SimCLR encompasses the entire unlabeled dataset, providing a reflection of data dispersion within the manifold.

SwAV also benefits from the clustering tendencies within the latent space to shape feature representations. The latent point cloud ($Z$) in SwAV is also constructed to capture the dispersion of data within the manifold, encompassing the entire unlabeled dataset. SwAV's key innovation compared to SimCLR is its shift from traditional contrastive learning to clustering-based learning, with the goal

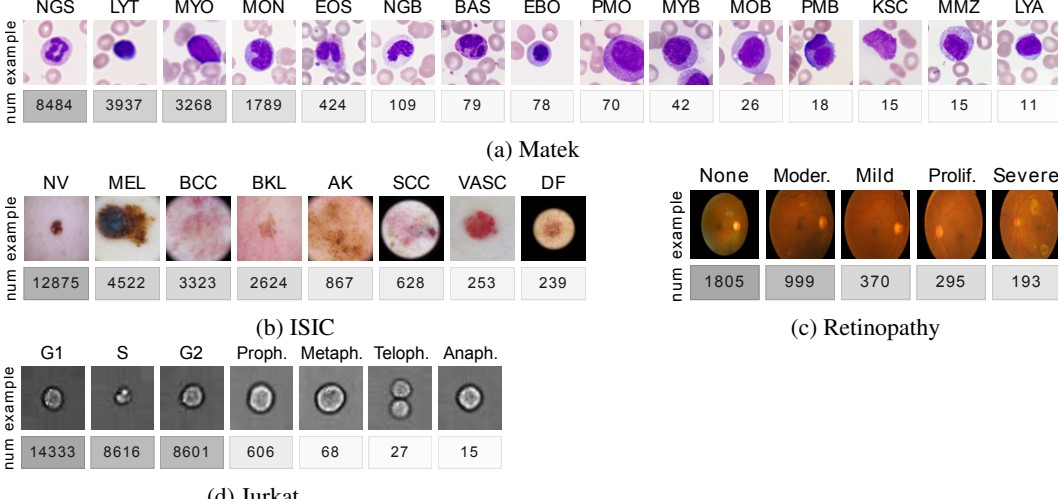

Figure 2: We benchmark various pretraining and sampling methods on four distinct biomedical datasets: (a) microscopic images of single white blood cells (Matek, $n = 18,365$), (b) skin lesion photographs (ISIC, $n = 25,331$), (c) fundus images for diabetic retinopathy detection and severity classification (Retinopathy, $n = 3,662$), and (d) imaging flow cytometry images of cell stages (Jurkat, $n = 32,266$). For each class, we present an example image and the total number of images.

of creating clusters of semantically similar data points. The use of multiple views and assignment swapping further enhances SwAV's ability to learn meaningful representations in natural images.

In contrast, DINO introduces a unique paradigm through self-distillation, where a teacher network ($f_{\theta_t}$) guides the learning of a student network ($f_{\theta_s}$) without explicit labels. DINO employs a distillation loss that encourages the student network to approximate the teacher's predictions. Several techniques, including centering and sharpening, are introduced to prevent mode collapse and enhance learning, making DINO distinctive in its approach.

Prioritizing classification tasks that necessitate discriminative feature learning, we chose self-supervised learning over generative models like Masked Autoencoders (MAE) (He et al., 2022). This aligns with the findings by Chandra et al. (2021) and Shetab Boushehri et al. (2022), where the effectiveness of discriminative SSL in biomedical data is demonstrated. Moreover, methods like MAE's dependence on large-scale Vision Transformers Dosovitskiy et al. (2021) was impractical for our dataset size.

# 4 EXPERIMENTS

## 4.1 DATA

We conduct experiments on four biomedical image datasets (see Figure 2).

- Matek: Microscopic images of single-cell white blood cells for studying Acute Myeloid Leukemia (AML) featuring 18,365 images in 15 classes (Matek et al., 2019).
- ISIC: Skin lesion photographs, with a focus on melanoma-related cases, consisting of 25,331 images categorized into eight diagnostic classes (Codella et al., 2018).
- Retinopathy: Fundus images for diabetic retinopathy detection and severity classification, encompassing 3,662 retina images in five severity classes (Karthik & Dane).
- Jurkat: Imaging flow cytometry images capturing cells in different cell cycle stages, with 32,266 images categorized into seven cell cycle phase classes (Eulenberg et al., 2017).

To address data sparsity in certain classes (see Figure 2), we adopt an 9:1 data split. We employ the validation set for monitoring mode collapse in the pretraining phase. The training split is utilized to train the backbone using SSL. Subsequently, we select sample points from the training split and paired data with their respective labels for supervised training of the classifier head. All reported

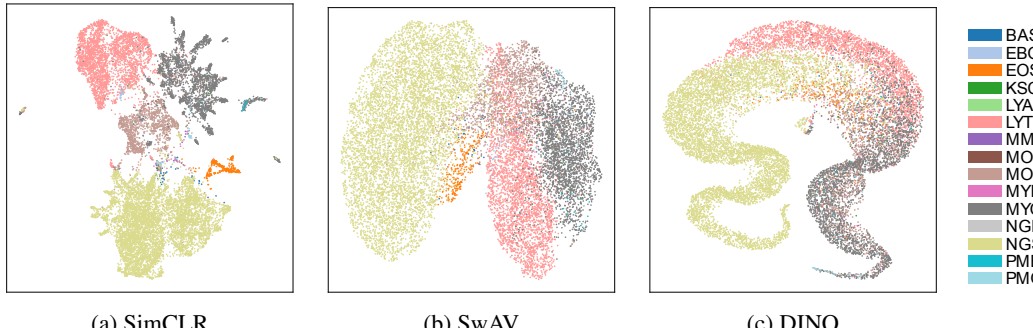

(a) SimCLR                    (b) SwAV                    (c) DINO

Figure 3: SimCLR outperforms other SSL techniques in generating a meaningful representation of unlabeled data. 2D UMAP representations of the latent space for the Matek dataset generated by (a) SimCLR, (b) SwAV, and (c)DINO. Cell types are shown in different colors. SimCLR excels by creating the most medically relevant clustered latent space, highlighting its effectiveness in capturing meaningful latent representations.

results are obtained from evaluations conducted on the isolated test split. To mitigate the influence of inherent randomness in our methods, we conduct each experiment five times, each time setting a different random seed. We report mean and standard deviation of the results.

## 4.2 TRAINING

We chose the Resnet-34 (He et al., 2016) encoder as the backbone $f_\theta(.)$ owing to its consistently robust performance across all datasets.

**Self-supervised pretraining.** We employ data augmentations from the original SimCLR method (Chen et al., 2020), such as random cropping, rotation, color jitter, and Gaussian blur. Similarly, we apply the original training configurations of DINO (Caron et al., 2021) and SwAV (Caron et al., 2020) for our experiments with these methods. These augmentations are adjusted for each dataset to ensure that augmented images did not become semantically ambiguous. Notably, we do not utilize the local views for DINO due to the use of relatively small image sizes (128 for Matek, ISIC, and Retinopathy, and 64 for Jurkat). To monitor the pretraining progress, we focus on detecting instances of mode collapse within the latent space. This is accomplished by observing the variance of image representations in the latent space during training.

SimCLR emerges as the standout performer, demonstrating its ability to handle datasets with high data imbalance. In DINO, which relies on self-distillation, the student network heavily relies on guidance from the teacher network. Consequently, when dealing with minority class data, it may face challenges in effectively distilling knowledge from the teacher network. SwAV, on the other hand, prioritizes the creation of well-balanced clusters as its primary objective. However, this emphasis on cluster balance may come at the cost of learning highly discriminative features, especially in complex datasets where subtle feature variations are essential for accurate classification or understanding. Furthermore, in imbalanced datasets where some classes have significantly fewer samples than others, clustering algorithms like SwAV may struggle to create well-balanced clusters, further complicating the learning process. To illustrate the performance of these three SSL techniques, we visualize 2D UMAP (Uniform Manifold Approximation and Projection), (McInnes et al., 2018) projection of Matek dataset in Figure 3.

**Sampling.** We conduct a comparative analysis with Cold PAWS (Mannix & Bondell, 2023), as the state-of-the-art method. We specifically consider the variant labeled as "best k-medoids (t-SNE)" as described in the original paper. Our implementation does not incorporate a semi-supervised approach on top of it; instead, we solely rely on labeled data for training the classifier head. We make this decision to ensure a fair comparison between models designed to accommodate unlabeled data and those that do not, considering the potential impact this might have on the results. We also train a classifier using images sampled entirely randomly for evaluation purposes. This random sampling approach establishes a robust baseline for our experiments, serving as a lower performance

bound for our proposed framework. It's worth mentioning that we refrain from comparing our work with Wang et al. (2022)'s approach due to the unavailability of their source code.

**Classifier head training.** We utilize a fixed pretrained backbone, denoted as $f_\theta$, to project each input image into a 512-dimensional feature space. On top of this feature representation, we train the classifier head with 512 input features and a number of output features, corresponding to the classes present within the annotation budget. Subsequently, we employ model soups to refine the classifier head weights. During testing, we utilize this trained model to classify data beyond the annotated subset. Any images belonging to classes not included in the subset are considered misclassified. To establish an upper performance bound, we train the classifier head with fully annotated data in a setting where proper validation is performed, focusing on maximizing the F1-macro score.

## 4.3 EVALUATION

**Performance.** Given the inherent data imbalance in all datasets, we assess the performance of our proposed method using various metrics, including F1-macro, balanced accuracy, Cohen's kappa, and the area under the precision-recall curve. Table 1, shows the performance of four samplings. We compare the results to random initialization (lower bound), supervised classifier training (upper bound) on the full data, and cold paws (Mannix & Bondell, 2023) to assess our approach's effectiveness in biomedical image processing. FPS outperforms other samplings in three of four datasets (for results on other metrics see Appendix A.1). The key insight from our table is that by judiciously selecting limited annotations, we can significantly narrow this gap from the initial state of active learning. This demonstrates that strategic data selection in the initial phase of active learning can approach the optimal performance achievable with a fully annotated dataset.

| Sampling method | F1-macro | | | | Balanced accuracy | | | |
|---|---|---|---|---|---|---|---|---|
| | Matek | ISIC | Retinopathy | Jurkat | Matek | ISIC | Retinopathy | Jurkat |
| Random | 0.30±0.01 | 0.30±0.02 | 0.46±0.04 | 0.23±0.01 | 0.32±0.02 | 0.34±0.03 | 0.47±0.04 | 0.25±0.02 |
| Cold paws | 0.37±0.02 | 0.30±0.02 | 0.49±0.04 | 0.23±0.01 | 0.42±0.04 | 0.33±0.02 | 0.50±0.04 | 0.24±0.03 |
| Furthest ($k$=100) | 0.37±0.02 | 0.30±0.02 | 0.50±0.03 | 0.21±0.03 | 0.43±0.04 | **0.36±0.02** | 0.52±0.04 | 0.31±0.08 |
| Closest ($k$=100) | 0.38±0.02 | 0.29±0.01 | 0.51±0.04 | 0.23±0.01 | 0.42±0.01 | 0.32±0.01 | 0.52±0.05 | 0.23±0.01 |
| Closest/furthest ($k$=50) | 0.38±0.02 | 0.31±0.01 | 0.50±0.04 | **0.24±0.01** | 0.43±0.03 | 0.35±0.02 | 0.50±0.05 | 0.28±0.01 |
| Furthest point sampling | **0.41±0.02** | **0.32±0.02** | **0.54±0.02** | 0.22±0.01 | **0.49±0.05** | 0.35±0.01 | **0.55±0.02** | **0.33±0.07** |
| Full data | 0.49±0.03 | 0.43±0.00 | 0.61±0.01 | 0.35±0.00 | 0.71±0.03 | 0.56±0.00 | 0.65±0.01 | 0.50±0.01 |

Table 1: FPS achieves the highest F1-macro score on Matek, ISIC, and Retinopathy, while for the Jurkat dataset closest/furthest sampling applied on $k$=50 clusters showed the best performance. The best performance is displayed in bold (excluding using full data). Mean and standard deviation is estimated for five runs for each experiment. $k$ in parentheses corresponds to the number of clusters in the pre-clustering step. Results show the performance of the classifier learned with 100 annotation budget.

**Class coverage.** Table 2 presents a comparative analysis of class coverage across our experiments. This assessment examines the ability of each approach to capture the diversity of classes within the dataset, particularly in scenarios involving class imbalances and critical classes. The table underscores the significance of sampling methods that demonstrate the best coverage across different data distributions. Our analysis shows how well each approach captures the diversity of classes within the dataset, which is crucial in imbalanced and critical-class scenarios. For instance, in the case of the retinopathy dataset, the latent distribution forms a clustered space where each cluster exhibits a heterogeneous distribution of all classes (see Appendix A.2). As a result, all sampling techniques excel in achieving optimal class coverage during initial data selection. Conversely, for Matek dataset, characterized by a high-class imbalance, features non-uniformly sized homogeneous clusters in the latent space (see Appendix 7). This poses a challenge for most sampling techniques to achieve comprehensive class coverage.

**Efficiency.** We assess the computational efficiency of different data sampling methods (Table 3). The the time complexity of FPS is $O(nm)$, while time complexity of Cold paws is $O(n^2 m)$, where $m$ represents the size of annotation budget and $n$ the size of whole dataset. Indeed, Cold paws, with its iterative t-SNE process, proves to be more computationally demanding, especially for large

| Sampling method | Matek | ISIC | Retinopathy | Jurkat |
|---:|:---:|:---:|:---:|:---:|
| Random | 5.6±0.9 | 7.4±0.5 | 5.0±0.0 | 4.0±0.7 |
| Cold paws | 8.2±1.3 | 6.8±0.4 | **5.0±0.0** | 4.0±1.0 |
| Furthest (k=100) | 10.0±0.7 | 7.6±0.5 | **5.0±0.0** | **5.8±0.4** |
| Closest (k=100) | 7.4±0.5 | 6.4±0.5 | **5.0±0.0** | 3.2±0.4 |
| Closest/furthest (k=50) | 8.4±0.9 | **7.8±0.4** | **5.0±0.0** | 4.4±0.5 |
| Furthest point sampling | **11.0±0.7** | **7.8±0.4** | **5.0±0.0** | 5.6±0.5 |
| Full data | 15.0±0.0 | 8.0±0.0 | 5.0±0.0 | 7.0±0.0 |

Table 2: Comparative class coverage analysis across diverse data distributions with annotation budget of 100. Over all datasets FPS provides the best coverage. All sampling techniques excel in the retinopathy dataset as classes are distributed in clusters (see Figure A.2). Values are mean and standard deviation from five independent runs.

datasets. In contrast, FPS, while slightly adding overhead compared to random sampling, remains highly efficient across various dataset sizes.

| Sampling method | Matek | ISIC | Retinopathy | Jurkat |
|---:|:---:|:---:|:---:|:---:|
| Random | 0.4±0.0 | 0.5±0.1 | 0.1±0.0 | 0.6±0.1 |
| Cold paws | 34.1±0.4 | 53.9±1.6 | 5.6±0.1 | 75.8±1.8 |
| Furthest (k=100) | 6.9±1.2 | 10.0±1.5 | 0.8±0.1 | 14.3±2.4 |
| Closest (k=100) | 7.7±1.4 | 10.3±2.0 | 0.7±0.0 | 11.8±1.1 |
| Closest/furthest (k=50) | 3.8±0.5 | 5.2±0.8 | 0.5±0.0 | 7.3±1.4 |
| Furthest point sampling | 6.8±1.4 | 8.6±1.7 | 0.7±0.3 | 10.0±0.3 |

Table 3: Sampling method runtimes in seconds ($\downarrow$) for an annotation budget of 100. Mean and standard deviation are calculated from 5 runs for each experiment. The worst-performing method is, cold paws (underlined). FPS is five times faster than this state-of-the-art.

**Ablation study.** We study the performance of all sampling strategies while we determine the optimal number of k-means clusters for each method. We widely experiment with the effect of k-mean clustering with a variant number of clusters on sampling performance (Figure 4). To see more exploration on sampling strategies using variant budgets and number of clusters please refer to Appendix A.3. We conduct similar experiments on bigger annotation budget (200, and 500 images). As expected, we observe a diminishing performance gap between our best-performing method and the random baseline, particularly as the annotation budget increases (Appendix A.4).

To evaluate the performance of different SSL techniques in our framework, we monitored the classification performance achieved base on each approach. Figure 5a shows F1-score for the FPS strategy, while Figure 5b illustrates the best classification outcomes with different sampling techniques applied to latent representations generated by three SSL techniques.

It's worth noting that FPS appears to share a conceptual similarity with algorithms like Gon algorithm (Dyer & Frieze, 1985). While Cold PAWS reports promising results with their approach, our experiments on biomedical datasets did not corroborate these findings. It's important to consider that Cold paws utilizes the testing dataset for early stopping, potentially introducing information leakage into their results.

**Model soups.** Figure 6 visually illustrates the effectiveness of model soups technique in enhancing the performance of our proposed method when a validation set is unavailable. The effect of model soups on the performance of our framework using other sampling techniques is shown in Appendix A.5.

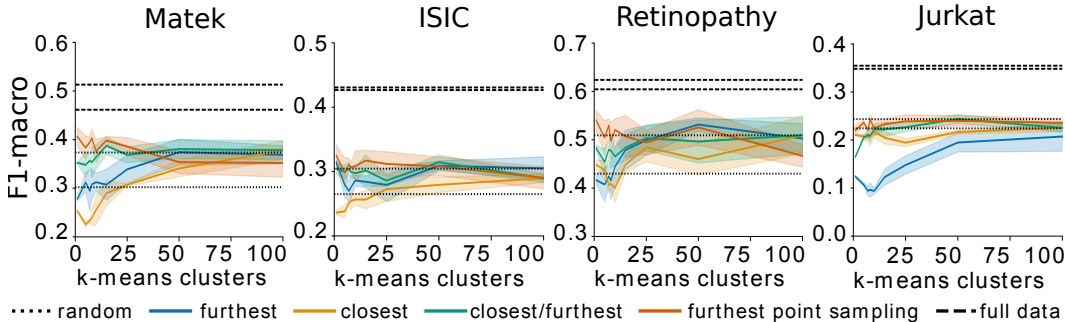

Figure 4: Consistant outperformance of FPS across all datasets compared to other sampling strategies, demonstrating superior performance without any need for clustering. Each case evaluates the effectiveness of different sampling strategies applied with varying numbers of clusters. Annotation budget is fixed to 100. Upper and lower bounds from five runs are shown with doted/dashed lines.

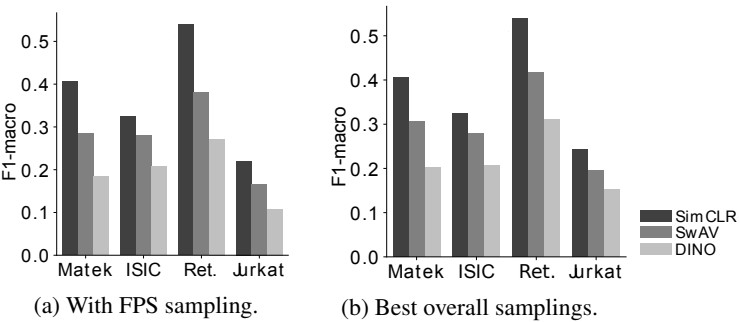

(a) With FPS sampling.      (b) Best overall samplings.

Figure 5: SimCLR provides the best latent representation for the cold start problem in biomedical datasets. We calculate F1-macro (↑) classification performance when the backbone is pretrained by different SSL techniques. (a) shows the performance based on FPS sampling, while (b) shows the best performance utilizing all sampling strategies (see also Appendix A.3).

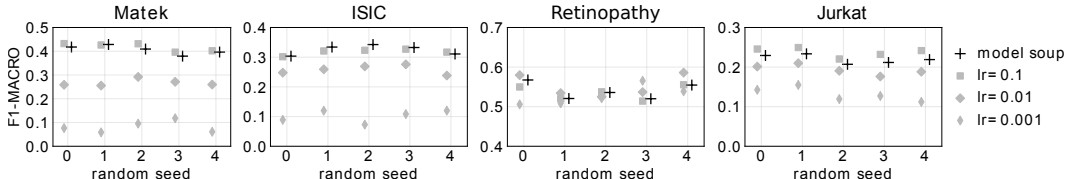

Figure 6: Optimized classifier parameters with model soups in absence of validation data. Our experiments utilize 100 labeled samples and multiple random seeds for each biomedical dataset and involve training three linear heads with varying learning rates (0.1, 0.01, and 0.001).

## 5 CONCLUSION

Our work proposes an effective solution to tackle the cold start problem in challenging biomedical datasets that excels in the absence of prior data knowledge or a suitable validation set. Our approach encompasses generating meaningful representations of unlabeled data, conducting diverse sampling while taking into account the data distribution density, and aggregating the optimal model weights even when a validation set is unavailable. Our work is a significant step towards the efficient annotation of unlabelled data. This is particularly relevant for the the development of decision support systems in medical diagnostics, where the annotation of large data sets is typically limited by expensiveness and scarcity of medical experts.

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

# A APPENDIX

## A.1 EVALUATION OF CLASSIFICATION PERFORMANCE ON OTHER METRICS

In this section, we present detailed results of our approach (see Table 4). Throughout the thesis, we utilized the F1-macro metric as our primary measure due to its sensitivity to classes regardless of their sample counts. Our choice of this metric is driven by the equal importance of all the classes in each dataset. Furthermore, to underscore the robustness of our approach, we offer additional widely-used metrics.

| Sampling method | Cohen's kappa (↑) | | | | Area under the precision-recall curve (↑) | | | |
|---|---|---|---|---|---|---|---|---|
| | Matek | Isic | Retinopathy | Jurkat | Matek | Isic | Retinopathy | Jurkat |
| random | 0.88±0.03 | 0.33±0.02 | 0.52±0.02 | 0.30±0.02 | 0.31±0.02 | 0.28±0.02 | 0.49±0.03 | 0.25±0.01 |
| cold paws | 0.86±0.05 | 0.35±0.03 | 0.57±0.03 | 0.28±0.03 | 0.40±0.01 | 0.29±0.02 | 0.51±0.03 | 0.24±0.01 |
| furthest (k=100) | 0.84±0.02 | 0.34±0.03 | 0.56±0.02 | 0.18±0.06 | 0.39±0.01 | 0.30±0.01 | 0.53±0.03 | 0.27±0.06 |
| closest (k=100) | **0.89±0.03** | 0.35±0.01 | 0.58±0.02 | **0.30±0.03** | 0.38±0.01 | 0.28±0.00 | 0.55±0.04 | 0.23±0.01 |
| closest/furthest (k=50) | 0.88±0.02 | 0.36±0.02 | 0.58±0.03 | 0.25±0.03 | 0.39±0.01 | 0.30±0.01 | 0.53±0.04 | 0.26±0.01 |
| furthest point sampling | 0.87±0.02 | **0.37±0.02** | **0.61±0.02** | 0.23±0.02 | **0.43±0.01** | **0.31±0.02** | **0.59±0.01** | **0.28±0.04** |
| full data | 0.80±0.03 | 0.46±0.00 | 0.65±0.01 | 0.47±0.00 | 0.58±0.02 | 0.49±0.00 | 0.68±0.01 | 0.46±0.01 |

Table 4: Other classification metrics (↑) for the annotation budget of 100. The best performance is always displayed in bold (excluding using full data).

## A.2 LATENT REPRESENTATION OF BIOMEDICAL DATASETS USING SSL TECHNIQUES

Figure7 shows the 2D UMAP projection of biomedical dataset's latent space achieved by three SSL techniques. Table 5

The strength of SimCLR compared to SwAV and DINO is shown in Table 5.

| Sampling method | SwAV | | | | DINO | | | |
|---|---|---|---|---|---|---|---|---|
| | Matek | Isic | Retinopathy | Jurkat | Matek | Isic | Retinopathy | Jurkat |
| random | 0.27±0.01 | 0.26±0.02 | 0.39±0.02 | 0.17±0.00 | 0.20±0.01 | 0.20±0.02 | **0.36±0.03** | 0.13±0.01 |
| cold paws | 0.28±0.01 | 0.27±0.02 | 0.39±0.03 | 0.16±0.01 | 0.17±0.03 | **0.21±0.01** | 0.29±0.04 | 0.14±0.00 |
| furthest (k=100) | **0.31±0.04** | 0.26±0.03 | 0.38±0.02 | **0.20±0.04** | 0.17±0.02 | 0.19±0.01 | 0.29±0.02 | 0.13±0.01 |
| closest (k=100) | 0.29±0.01 | 0.25±0.01 | **0.42±0.03** | 0.15±0.01 | 0.20±0.01 | 0.19±0.02 | 0.31±0.04 | 0.15±0.00 |
| closest/furthest (k=50) | 0.29±0.01 | 0.25±0.02 | 0.39±0.03 | 0.19±0.03 | **0.20±0.02** | 0.17±0.00 | 0.28±0.04 | **0.15±0.01** |
| furthest point sampling | 0.29±0.02 | **0.28±0.03** | 0.38±0.02 | 0.17±0.04 | 0.18±0.03 | **0.21±0.01** | 0.27±0.01 | 0.11±0.01 |
| full data | 0.47±0.01 | 0.38±0.00 | 0.46±0.01 | 0.32±0.00 | 0.24±0.00 | 0.26±0.00 | 0.37±0.01 | 0.21±0.01 |

Table 5: Other self-supervised learning techniques. We observe that the performance benefit of our non-naive methods over the random baseline is less prominent (or worse) with both self-supervised methods.

## A.3 SAMPLING ON TOY DATASETS

Here we illustrate the coverage of various sampling techniques across different data distribution models, as explored in our ablation study. In this figure, each column shows a toy distribution, and each row depicts the result of each sampling strategy using hyperparameters of their best performance estimated with our main experiments. Additionally, we include the results obtained using the cold PAWS method for comprehensive comparison. For visual demonstrations of the tested sampling methods on toy datasets generated from typical probability distributions, see Figure 8).

## A.4 DETAILED INFLUENCE OF ANNOTATION BUDGET SIZE ON THE PERFORMANCE

To gain a deeper understanding of the scalability and potential of our approach, we executed experiments with varying initial annotation budgets. Specifically, we explored annotation budgets of

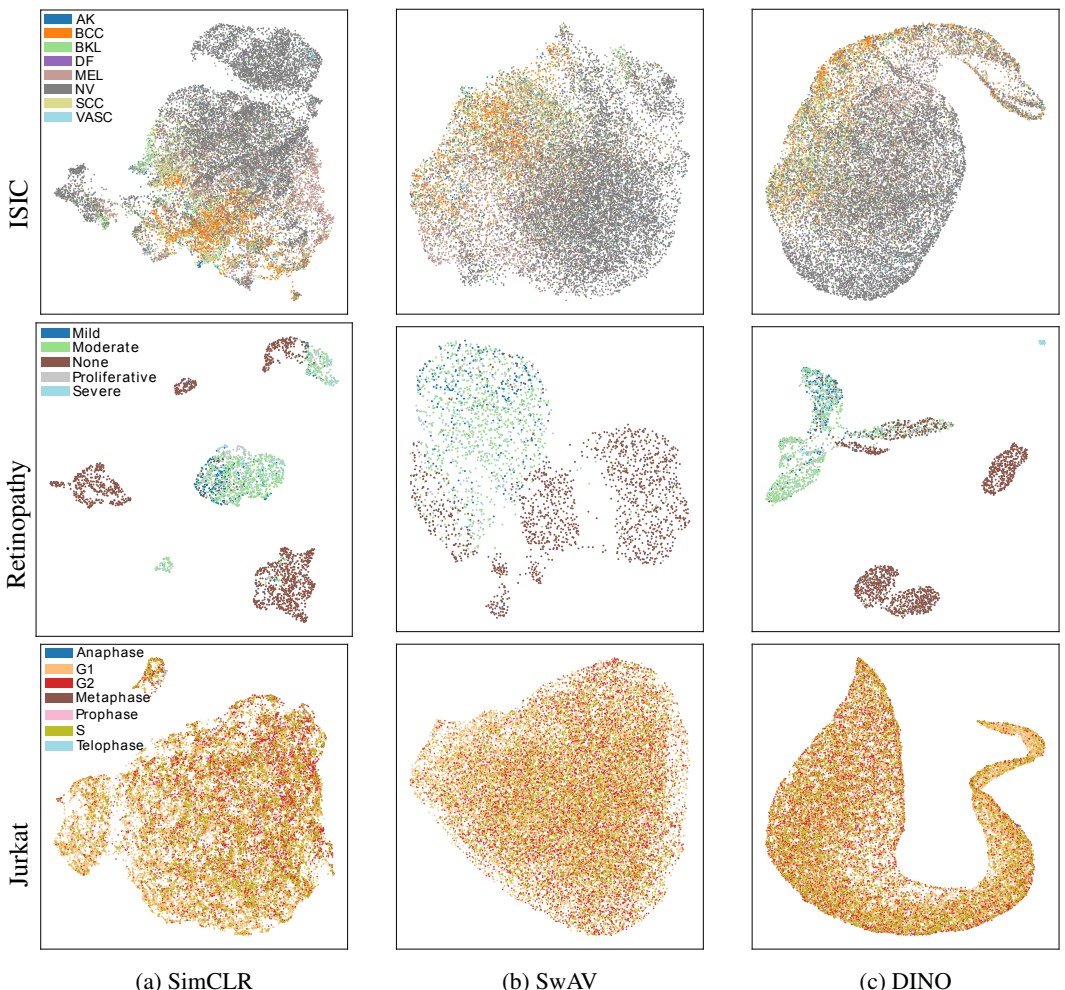

(a) SimCLR      (b) SwAV      (c) DINO

Figure 7: Biomedical Data Latent Distribution is shown here. In these UMAP 2D representations of four biomedical datasets, (a) Matek, (b) ISIC, (c) Retinopathy, and (d) Jurkat, colored by class, we visualize the latent distribution created by SimCLR, SwAV, and DINO (rows up-down). Matek exhibits a homogeneously clustered class distribution, whereas Retinopathy displays a more heterogeneous distribution. Jurkat and ISIC datasets do not exhibit strong clustering.

100, 200, and 500 images (representing less than 3%, 7%, and 17% of the smallest dataset (i.e., Retinopathy)), shedding light on the utility of our method as the annotation budget increases (see Table 6 Figure 9).

| Sampling method | annotation budget of 200 | | | | annotation budget of 500 | | | |
|---|---|---|---|---|---|---|---|---|
| | Matek | Isic | Retinopathy | Jurkat | Matek | Isic | Retinopathy | Jurkat |
| random | 0.33±0.02 | 0.32±0.02 | 0.51±0.03 | 0.25±0.01 | 0.37±0.04 | 0.32±0.01 | 0.57±0.05 | 0.26±0.01 |
| cold paws | 0.38±0.01 | 0.32±0.00 | 0.54±0.02 | 0.24±0.01 | 0.40±0.02 | 0.32±0.02 | 0.54±0.03 | 0.25±0.01 |
| furthest (k=200\|500) | 0.35±0.01 | 0.32±0.01 | 0.51±0.03 | 0.22±0.01 | 0.37±0.02 | 0.33±0.00 | 0.56±0.02 | 0.26±0.01 |
| closest (k=200\|500) | 0.40±0.01 | 0.30±0.01 | 0.52±0.02 | 0.25±0.00 | **0.41±0.00** | 0.33±0.01 | 0.57±0.03 | 0.26±0.01 |
| closest/furthest (k=100\|250) | 0.39±0.02 | 0.33±0.01 | 0.54±0.03 | **0.26±0.02** | 0.40±0.01 | 0.33±0.00 | 0.57±0.01 | **0.28±0.01** |
| furthest point sampling | **0.42±0.01** | **0.34±0.01** | **0.55±0.03** | 0.24±0.01 | **0.41±0.00** | **0.35±0.01** | **0.60±0.03** | 0.26±0.01 |
| full data | 0.49±0.03 | 0.43±0.00 | 0.61±0.01 | 0.35±0.00 | 0.49±0.03 | 0.43±0.00 | 0.61±0.01 | 0.35±0.00 |

Table 6: F1-macro (↑) for the annotation budget of 200 and 500.

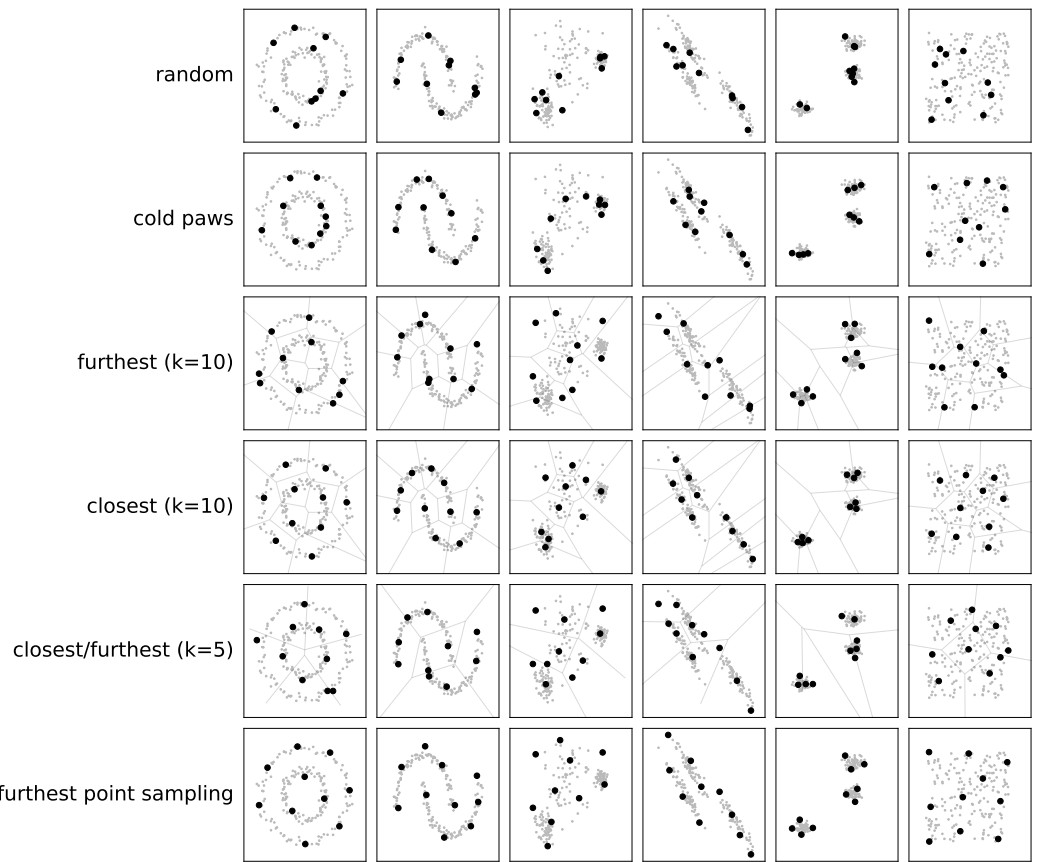

Figure 8: Visual demonstration of different sampling methods. Points selected by a given method are drawn in black. Where preclustering was used, we also provided separation boundaries between neighboring clusters.

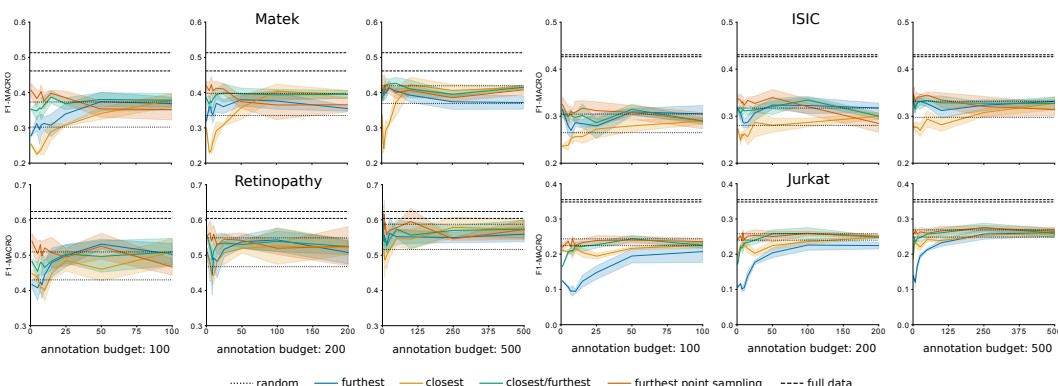

Figure 9: Visual demonstration of different sampling methods. Points selected by a given method are drawn in black. Where preclustering was used, we also provided separation boundaries between neighboring clusters.

## A.5 MODEL SOUPS VS. SAMPLING

Model soups, in general, emerged as a crucial strategy, particularly for enhancing robustness and facilitating domain knowledge transfer. Demonstrating their effectiveness, we empirically show

how averaging weights of models (i.e., uniform soup, as referred to in the original paper) trained with varying hyperparameters—such as learning rates in our case—can circumvent the need for validation while yielding comparable results (see Figure 10).

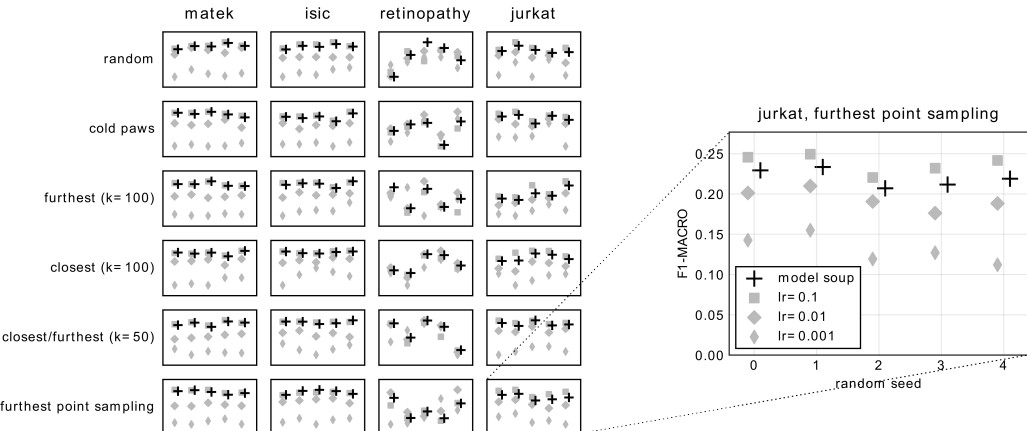

Figure 10: Comprehensive comparison of model soups performance against varying learning rates across multiple datasets and sampling strategies. The datasets are represented column-wise, while the different sampling techniques are delineated row-wise. Within each experimental setup, we trained the classifier head with 5 distinct initial weight configurations (depicted on the x-axis) combined with various learning rates (indicated by different shapes in the plot). The results consistently demonstrate that model soups, in most instances, effectively match or surpass the performance achieved by the most optimal learning rate. Results are shown for the annotation budget of 100.

