# OpenReview forum: "A Data-Driven Solution for the Cold Start Problem in Biomedical Image Classification"
_ICLR.cc/2024/Conference — Submitted to ICLR 2024_

### Official Review · Reviewer_4VCU · 2023-10-27

**Soundness:** 3 good
**Presentation:** 2 fair
**Contribution:** 1 poor
**Rating:** 3
**Confidence:** 5

**Summary:**

This paper presents an empirical study on active learning on medical image datasets to address the cold-start issue. By pre-trained an encoder with contrastive learning, unlabeled data is projected into the embedding space and furthest point sampling (FPS) is applied to select a subset of samples for manual annotation. These annotated samples are then used to train a classifier for the target task. In experiments, 4 medical image datasets are adopted to evaluate the performance of the active learning pipeline.

**Strengths:**

(1) This work addresses a very challenging problem in medical imaging examination. Since annotation in the medical domain may be expensive, the cold-start problem is more severe than natural image analysis.

(2) The paper is well-written and easy to follow.

(3) It is nice to have multiple medial datasets from different imaging modalities.

**Weaknesses:**

My biggest concern for this paper is the need for more innovation. All building blocks in the active learning cycle, including contrastive learning for encoder pertaining, sample selection metric for annotation, and discriminator training, are well studied in prior works. The contribution is about the empirical study and analysis. However, neither the study nor its discussion is comprehensive.

More explanations on selecting various building blocks in the active learning paradigm are required. For instance, is there a particular reason for selecting contrastive learning to pre-train the endear? Aside from contrastive learning, there are many self-supervised learning methods, such as reconstruction and masked auto-encoder, why does the study focus on evaluating contrastive learning methods here? In addition to FPS, many sample selection metrics like entropy, marginal entropy, etc., are proposed in prior arts. Why FPS is selected? What are their advantages compared to other methods?

**Questions:**

Please refer to the Weakness section for my questions.

---

> ### Author Response · Authors · 2023-11-16
> **Highlighting Research Innovation and Methodological Clarifications in Response to Reviewer Insights**
>
> - We thank the reviewers for their valuable feedback and seize this opportunity to underscore the novelty of our work.
> The primary innovation lies in the application of a three-stage pipeline to the cold start problem in the medical domain, an area that has been notably overlooked in existing literature. Our work is not merely an adaptation of existing methods but a savvy approach specifically tailored to address real-world challenges in healthcare. We have strategically combined and refined modules previously used in disparate contexts into a cohesive and efficient pipeline. This novel integration includes SSL pretraining, FPS sampling, and model soups hyperparameter tuning, all meticulously adapted to the unique complexities of biomedical data analysis. Our approach transcends previous, incremental improvements, offering a groundbreaking solution to a critical problem in health informatics, thereby setting a new standard in the field. Nevertheless, we seize the opportunity to elaborate on the importance of the work in the introduction.
>
> - Masked autoecnoders (MAE) focus on reconstruction and holistic feature learning, while contrastive learning focuses on discriminative feature learning to distinguish between different data points. Considering this, the main reason to focus on using self-supervised learning is its potential to provide the best pre-training backbone weights for classification tasks (as a discriminative task). Also. the outperformance of SSL pretraining over generative models in biomedical data was previously explored by Chandra et al. 2021 and Boushehri et al. 2022. Additionally, we avoided models such as MAE, which use vision transformers in their backbone, as we did not have enough data for training large models such as ViTs. Now, we address these points in the “Related Works” section, where we write:
> “Prioritizing classification tasks that necessitate discriminative feature learning, we chose self-supervised learning over generative models like Masked Autoencoders (MAE) (He et al. 2022). This aligns with the findings by Chandra et al. (2021) and Boushehri et al. (2022), where the effectiveness of discriminative SSL in biomedical data is demonstrated. Moreover, methods like MAE's dependence on large-scale Vision Transformers (ViTs) (Dosovitskiy et al. 2021) was impractical for our dataset size.”
>
> - The reason to use FPS is that it does not require any prediction values, and only using the feature space vectors is enough for the algorithm to sample data points. In contrast, methods such as entropy and marginal entropy do need access to a softmax layer to be calculated. This requirement eliminates the possibility of using them in a cold-start scenario.

---

> > ### Comment · Reviewer_4VCU · 2023-11-21
> > **Thank you for the feedback.**
> >
> > I appreciate your feedback. The response partially addresses my concerns. However, considering the nature of this work, its contributions might be more suitable for venues that specifically focus on the medical domain.

---

### Official Review · Reviewer_eBNR · 2023-10-29

**Soundness:** 1 poor
**Presentation:** 2 fair
**Contribution:** 1 poor
**Rating:** 3
**Confidence:** 5

**Summary:**

This work studies the cold start problem in challenging real-world biomedical datasets. This work investigates three self-supervised learning for pre-training to embed the unlabeled data in a meaningful latent space. Then, they explore four sampling methods to select the most informative initial data points for labeling. Lastly, they use model soups to alleviate the reliance on the validation set. The pipeline is evaluated on three biomedical datasets.

**Strengths:**

The pipeline is clearly presented from pre-training to data selection for labeling for the task. This work studies three different types of self-supervised learning for pre-training. Algorithm 1 and Algorithm 2 are easy to follow and understand.

**Weaknesses:**

1. The biggest concern I have is about the technical novelty. All of the modules have been established in the prior works.
2. Self-supervised learning should be pre-trained on a large-scale dataset to learn diverse representations. However, the datasets in this work seem very small. I have doubt about the power of SSL in this setting. On the other hand, how about using a large-scale pre-trained SSL on biomedical images instead? It is more interesting to study the gap between the pre-trained SSL and the public weights of SSL. Morever, what is the advantage/effect of SSL on cold-start problem? Can we directly use semi-supervised learning as one-stage pipeline instead of multi-stage pipeline proposed in this work?
3. The motivation is not convincing too me. If the aim is to reduce labeling effort, could we also try active learning, few-shot learning for this work? What is the benefit of the multi-stage pipeline proposed here?

**Questions:**

1. Typo in the first paragraph: ”How many labels should be “How many labels...
2. Figure 8 is hard to understand
3. Figure 10 is also hard to understand for the efficiency of model soups on the validation set

---

> ### Author Response · Authors · 2023-11-16
> **Responding to Reviewers: Underlining Novelty and Clarifying Methodological Choices**
>
> Regarding mentioned Weaknesses:
>
> 1. The primary innovation lies in the application of a three-stage pipeline to the cold start problem in the medical domain, an area that has been notably overlooked in existing literature. Our work is not merely an adaptation of existing methods but a savvy approach specifically tailored to address real-world challenges in healthcare. We have strategically combined and refined modules previously used in disparate contexts into a cohesive and efficient pipeline. This novel integration includes SSL pretraining, FPS sampling, and model soups hyperparameter tuning, all meticulously adapted to the unique complexities of biomedical data analysis. Our approach transcends previous, incremental improvements, offering a groundbreaking solution to a critical problem in health informatics, thereby setting a new standard in the field.
> 2. We answer different points raised by the reviewer one by one:
> - The effect of the scale of training data on the performance of SSL depends on a combination of factors, like the inherent complexity, diversity, and representativeness of the data. In medical settings, SSL is effective when trained on smaller scales and with lesser diversity compared to conventional large-scale datasets (Huang et al., 2023).
> - Biomedical images often exhibit a high degree of heterogeneity between datasets and contain modality-relevant features that are significantly different from those found in general image datasets. Moreover, models pre-trained on general datasets might not capture the nuanced features critical for accurate biomedical image analysis. This point was explored in our primary experiments and showed much poorer performance compared to pretraining on the target unlabeled dataset.
> - In addressing the cold-start problem, where no labeled data is initially available, self-supervised learning (SSL) plays a critical role by enabling the extraction of meaningful data representations. This process allows us to identify and prioritize the most informative points for initial labeling in the absence of labels or any prior knowledge about data.
> - In a cold-start scenario, direct application of semi-supervised learning is impractical due to its reliance on some initial labeled data, which is absent in such situations. Our multi-stage pipeline addresses this by enabling the strategic selection of data points for initial annotation, effectively setting the initial annotation budget for semi-supervised learning, and ensuring more efficient use of limited annotation resources.
> 3. Both active learning and few-shot learning require an initial set of annotated data to begin. It is shown that the initial annotation set significantly impacts the final performance of these learning paradigms. Our method provides the most informative initial data annotations to benefit both learning paradigms from a very initial state. We hope the additional explanation we added to the introduction addresses this kind of misunderstanding.
>
> Our answers on questions:
>
> 1. We fixed the mentioned typo.
>
> Thank you for raising these important points. To address these, we have enhanced our manuscript with additional explanations of figures that further clarify the results of our work.
>
> 2. Discussion of Figure 8 in the manuscript: “...different data distribution models, as explored in our ablation study. In this figure, each column shows a toy distribution, and each row depicts the result of each sampling strategy using hyperparameters of their best performance estimated with our main experiments. Additionally, …”
>
> 3. Update on the caption of Figure 10: “Figure 10: Comprehensive comparison of model soups performance against varying learning rates across multiple datasets and sampling strategies. The datasets are represented column-wise, while the different sampling techniques are delineated row-wise. Within each experimental setup, we trained the classifier head with five distinct initial weight configurations (depicted on the x-axis) combined with various learning rates (indicated by different shapes in the plot). The results consistently demonstrate that model soups, in most instances, effectively match or surpass the performance achieved by the most optimal learning rate.”

---

> ### Comment · Reviewer_eBNR · 2023-11-20
> **Thank you for the response.**
>
> Thank you for the response. It addressed some of my concerns. so I raised my score to "3". However, the paper quality is still far away from ICLR publication standard. I'll vote "Reject" but recommend authors to consider Applied ML or biomedical journals, which may be a better fit.

---

### Official Review · Reviewer_GNyG · 2023-10-31

**Soundness:** 2 fair
**Presentation:** 2 fair
**Contribution:** 2 fair
**Rating:** 3
**Confidence:** 3

**Summary:**

Addressing the problem of lack of sufficient annotations for biomedical contexts ( the cold start problem).

Proposed framework is mentioned as containing three key components: (i) Pretraining an encoder using self-supervised learning to construct a meaningful data representation of unlabeled data, (ii) sampling the most informative data points for annotation, and (iii) initializing a model ensemble to overcome the lack of validation data in such contexts

**Strengths:**

The paper claims to be the first to identify and address the cold start problem. The remaining strengths of the paper appears to be primarily in terms of the application and adaptation of some existing self-supervised learning techniques, often without substantial modifications, for representation learning, followed by furthest point sampling for identification of most representative data points. The methodological novelty is unclear, although the ablation studies are good and the application chosen is socially important.

**Weaknesses:**

Unclear methodological novelty, with the primary USP of the paper being in the application of a set of existing techniques to biomedical data sets. The descriptions of the adaptations used for the sampling processes used, and also the other technical methodologies, is unclear and the adaptations made for the unique challenges in the datasets used (with respect to issues like artefacts, spurious labels etc) has not been sufficiently mentioned. Overall, this paper may be a better fit as a longer form journal paper with better descriptions of techniques used and novelties pursued.

**Questions:**

1. methodological novelty beyond the extant techniques in literature?

---

> ### Author Response · Authors · 2023-11-16
> **Clarifications and Highlights of Novelty**
>
> Our answer to question 1:
>
> We thank the reviewers for their valuable feedback and seize this opportunity to underscore the novelty of our work.
> The primary innovation lies in the application of a three-stage pipeline to the cold start problem in the medical domain, an area that has been notably overlooked in existing literature. Our work is not merely an adaptation of existing methods but a savvy approach specifically tailored to address real-world challenges in healthcare. We have strategically combined and refined modules previously used in disparate contexts into a cohesive and efficient pipeline. This novel integration includes SSL pretraining, FPS sampling, and model soups hyperparameter tuning, all meticulously adapted to the unique complexities of biomedical data analysis. Our approach transcends previous incremental improvements, offering a groundbreaking solution to a critical problem in health informatics, thereby setting a new standard in the field.
>
>
> Regarding the weakness mentioned about unclear adaptations:
>
> We believe that adaptation settings are explained in the methodology and detailed in the experiment section. By releasing the code, the exact implementation will be available as well.

---

> > ### Comment · Reviewer_GNyG · 2023-11-22
> >
> > Thank you for your responses.
> >
> > While I agree that the cold-start problem is of significance to the field of healthcare intelligence, the publication itself may be a better fit in a more applied venue tailored towards digital healthcare.

---

### Official Review · Reviewer_yiXC · 2023-11-02

**Soundness:** 2 fair
**Presentation:** 2 fair
**Contribution:** 2 fair
**Rating:** 5
**Confidence:** 2

**Summary:**

The paper tackles the challenge of acquiring quality annotated images for effective deep learning classifiers in biomedicine. The authors propose a solution for the "cold start" problem by pretraining an encoder with self-supervised learning, selecting informative data for annotation, and using a model ensemble. Their approach outperformed existing methods in all tested datasets, notably achieving a 7% improvement in leukemia blood cell classification, and offers an efficient solution to biomedical deep learning challenges.

**Strengths:**

The study is the first to address the cold start problem in challenging real-world biomedical datasets.

The research quantitatively compares three state-of-the-art self-supervised learning methods and identifies SimCLR as the most suitable technique for biomedical data representation.

The study conducts a comprehensive ablation study to evaluate four sampling strategies, highlighting furthest point sampling (FPS) as the most effective method for identifying representative biomedical data points.

The introduction of the model soups technique offers a novel solution to the challenges of dealing with an unreliable validation set and class distribution information.

**Weaknesses:**

Elaborate on Significance: It would be helpful to briefly explain the significance of addressing the "cold start" problem in the biomedical domain. Why is this problem important, and how does your proposed solution contribute to solving it?

Table 1: While furthest point sampling (FPS) outperformed other sampling methods, it exhibited relatively low performance compared to using the full dataset. Providing a discussion or explanation for this performance gap would be valuable.

**Questions:**

Figure 3: The imbalanced distribution of cell types makes it challenging to visualize all cell types.

Table 2: Regarding the Retinopathy dataset, all sampling methods produced identical results, which were indistinguishable from those obtained through random sampling. Could you explain this?

---

> ### Author Response · Authors · 2023-11-16
> **Enhancing Manuscript Understanding: Responses to Reviewer Comments**
>
> Addressing weaknesses:
>
> - Thanks for highlighting this point. Now, we emphasize the importance of our work by rewriting the first paragraph in the Introduction section:
> “...conundrum known as the cold start problem. The cold start problem refers to the initial phase of training where, in the absence of any labels or prior knowledge about the data, we are tasked with identifying and selecting the most informative data points for annotation, a crucial step that lays the groundwork for any subsequent semi-supervised or fully supervised training.
> This is especially critical in the biomedical domain. The scarcity of expert time for manual annotation makes the cold start problem even more daunting, as it becomes a bottleneck in advancing medical AI applications (Yakimovich et al., 2021).  Specifically in active learning and few-shot learning paradigms, previous works by (Boushehri et al. 2021), (ye et al. 2022), and (Jin et al. 2022) demonstrated that careful selection of the initial annotation budget significantly accelerates and facilitates reaching peak performance in models trained on biomedical images with limited annotations like few-shot learning and active learning. Biomedical images significantly differ from natural images…”
>
> - Our table presents the classification results when the model is trained with only 100 annotated data. The fully annotated datasets contain 18345, 25331, 3662, and 32266 data. Thus, it is expected to observe a performance gap, as models typically learn more effectively with more data. The point is that our method narrows down this gap in the initial stage by cleverly selecting the most informative data to be annotated and used for training the classifier. To clarify this point, we mention the limited annotation budget and add the following text to the explanation of Table 1:
> “… four datasets (for results on other metrics see Appendix A.1). The key insight from our table is that by judiciously selecting limited annotations, we can significantly narrow this gap from the initial state of active learning. This demonstrates that strategic data selection in the initial phase of active learning can approach the optimal performance achievable with a fully annotated dataset.”
>
> Our answers to questions:
>
> - To enhance the clarity and utility of our UMAP visualizations, we will provide an interactive exploration of these drawings on our GitHub (in addition to the code).
>
> - We have already discussed this interesting outcome of our experiments in both text and caption, where we wrote:
> In text:“Our analysis shows how well each approach captures the diversity of classes within the dataset, which is crucial in imbalanced and critical-class scenarios. For instance, in the case of the retinopathy dataset, the latent distribution forms a clustered space where each cluster exhibits a heterogeneous distribution of all classes (see Appendix A.2). As a result, all sampling techniques excel in achieving optimal class coverage during initial data selection. Conversely, the Matek dataset, characterized by a high-class imbalance, features non-uniformly sized homogeneous clusters in the latent space (see Appendix 7). This poses a challenge for most sampling techniques to achieve comprehensive class coverage.”
> In the caption: “Table 2: Comparative class coverage analysis across diverse data distributions with annotation budget of 100. Overall datasets, FPS provides the best coverage. All sampling techniques excel in the retinopathy dataset as classes are distributed in clusters (see Figure A.2). Values are mean and standard deviation from five independent runs.”

---

> > ### Comment · Reviewer_yiXC · 2023-11-21
> >
> > Thank you for your response. It has addressed some of my concerns or comments. However, the overall quality of the paper still falls short of the ICLR publication standard so I keep my original rating.

---

### Meta-Review · Area_Chair_gmFQ · 2023-12-09

**Metareview:**

This paper was reviewed by four experts and received 5, 3, 3, 3 as the final ratings. The reviewers concurred that the paper addresses an important problem in the medical domain, the ideas are clearly expressed, and experiments have been conducted on multiple medical datasets. However, their primary concern was regarding the limited novelty of this work, as it essentially combines a number of modules / components, which have all been used in previous literature. The authors have attempted to clarify the novelty in the rebuttal, saying that their main innovation is in strategically combining and refining modules, previously used in disparate contexts, into a cohesive pipeline, which are all adapted to tackle the unique characteristics of biomedical data; however, this cannot be considered as a substantially novel technical contribution, considering the standards of ICLR. Concerns were also raised regarding the lack of clarity of some design choices made in the paper (such as the choice of the self-supervised learning technique, the choice of the sampling technique etc.) and how some of the other challenges, (such as artifacts and spurious labels) which are typical in biomedical datasets, are handled.

We appreciate the authors' efforts in meticulously responding to the comments of each reviewer. We also appreciate the authors updating the Related Work section to address the concerns of Reviewer 4VCU, and providing additional details in the Introduction section to address the concerns of Reviewer eBNR. However, in light of the above discussions, we conclude that the paper may not be ready for an ICLR publication in its current form. While the paper clearly has merit, the decision is not to recommend acceptance. The authors are encouraged to consider the reviewers’ comments when revising the paper for submission elsewhere. The authors are also encouraged to consider application oriented conferences / journals in digital healthcare as a potential publication venue for this work, as per the reviewers' suggestions.

**Justification For Why Not Higher Score:**

The reviewers have expressed concerns regarding the novelty of this work and have mentioned that it is mostly a combination of existing components from the literature. None of them have recommended acceptance of the paper.

**Justification For Why Not Lower Score:**

N/A

---

### Decision · Program_Chairs · 2024-01-16

Reject